# Extension of Lung Damage at Chest Computed Tomography in Severely Ill COVID-19 Patients Treated with Interleukin-6 Receptor Blockers Correlates with Inflammatory Cytokines Production and Prognosis

Lucio Calandriello [1,†], Enrico De Lorenzis [2,†], Giuseppe Cicchetti [1], Rosa D'Abronzo [1], Amato Infante [1], Federico Castaldo [1], Annemilia Del Ciello [1], Alessandra Farchione [1], Elisa Gremese [3,4], Riccardo Marano [1,5], Luigi Natale [1,5], Maria Antonietta D'Agostino [2,4], Silvia Laura Bosello [2,4,*] and Anna Rita Larici [1,5]

1   Department of Diagnostic Imaging, Oncological Radiotherapy and Hematology, Diagnostic Imaging Area, Fondazione Policlinico Universitario Agostino Gemelli, IRCCS, L.go Agostino Gemelli 8, 00168 Rome, Italy; lucio.calandriello@policlinicogemelli.it (L.C.); giuseppe.cicchetti@policlinicogemelli.it (G.C.); rosa.dabronzo2@guest.policlinicogemelli.it (R.D.); amato.infante@policlinicogemelli.it (A.I.); federicocastaldo@ymail.com (F.C.); annemilia.delciello@policlinicogemelli.it (A.D.C.); alessandra.farchione@policlinicogemelli.it (A.F.); riccardo.marano@policlinicogemelli.it (R.M.); luigi.natale@policlinicogemelli.it (L.N.); annarita.larici@policlinicogemelli.it (A.R.L.)
2   Unit of Rheumatology, Fondazione Policlinico Universitario Agostino Gemelli IRCCS, L.go Agostino Gemelli 8, 00168 Rome, Italy; delorenzis.e@gmail.com (E.D.L.); mariaantonietta.dagostino@policlinicogemelli.it (M.A.D.)
3   Division of Clinical Immunology, Fondazione Policlinico Universitario Agostino Gemelli IRCCS, L.go Agostino Gemelli 8, 00168 Rome, Italy; elisa.gremese@policlinicogemelli.it
4   Department of Geriatric and Orthopaedic Sciences, Università Cattolica del Sacro Cuore, L.go Francesco Vito 1, 00168 Rome, Italy
5   Department of Radiological and Hematological Sciences, Section of Radiology, Università Cattolica del Sacro Cuore, L.go Francesco Vito 1, 00168 Rome, Italy
*   Correspondence: silvialaura.bosello@policlinicogemelli.it; Tel.: +39-3348415847
†   These authors contributed equally to this work.

**Abstract:** Elevated inflammatory markers are associated with severe coronavirus disease 2019 (COVID-19), and some patients benefit from Interleukin (IL)-6 pathway inhibitors. Different chest computed tomography (CT) scoring systems have shown a prognostic value in COVID-19, but not specifically in anti-IL-6-treated patients at high risk of respiratory failure. We aimed to explore the relationship between baseline CT findings and inflammatory conditions and to evaluate the prognostic value of chest CT scores and laboratory findings in COVID-19 patients specifically treated with anti-IL-6. Baseline CT lung involvement was assessed in 51 hospitalized COVID-19 patients naive to glucocorticoids and other immunosuppressants using four CT scoring systems. CT data were correlated with systemic inflammation and 30-day prognosis after anti-IL-6 treatment. All the considered CT scores showed a negative correlation with pulmonary function and a positive one with C-reactive protein (CRP), IL-6, IL-8, and Tumor Necrosis Factor α (TNF-α) serum levels. All the performed scores were prognostic factors, but the disease extension assessed by the six-lung-zone CT score (S24) was the only independently associated with intensive care unit (ICU) admission ($p = 0.04$). In conclusion, CT involvement correlates with laboratory inflammation markers and is an independent prognostic factor in COVID-19 patients representing a further tool to implement prognostic stratification in hospitalized patients.

**Keywords:** computed tomography; COVID-19; prognosis; cytokine release syndrome; interleukin-6

## 1. Introduction

At the end of 2019, a novel coronavirus, designated as severe acute respiratory syndrome coronavirus 2 (SARS-CoV-2), became responsible for a pandemic [1]. Most cases of

COVID-19, the disease caused by SARS-CoV-2 infection, are mild or asymptomatic [2], with a minority of patients developing a severe or life-threatening infection characterized by an excessive cytokine release from abnormal and non-effective immune response [3]. This hyperimmune response, defined as a cytokine storm (CS), has been considered responsible for lung injury and widespread tissue damage [4]. CS occurs in several conditions, even though the way it presents in COVID-19 is unique and has not been clearly defined yet. Preliminary criteria for the prediction of CS in COVID-19 patients, based mainly on laboratory results, have been defined in order to identify patients with a more severe disease course [5].

Several anti-inflammatory and immunomodulatory drugs targeting cytokines have been tested to hinder the effect of severe COVID-19, producing conflicting results [6]. Interleukin-6 (IL-6) is one of the cytokines involved in SARS-CoV-2-induced inflammatory response and monoclonal antibodies against the interleukin-6 receptor (IL-6R)—namely tocilizumab (TCZ) and sarilumab (SAR)—have been used in severe infection [7,8]. In this scenario, it is important to identify biomarkers that can predict the prognosis, the onset of CS and the response to these kinds of immunomodulatory drugs. Of note, the prognostic role of laboratory biomarkers, particularly those related to systemic immune response, has been at least in part weakened in clinical practice because of the common use of glucocorticoids, even before hospital admission, in particular from the second pandemic wave onwards [9].

Chest imaging plays a pivotal role in the diagnosis of COVID-19, and findings associated with SARS-CoV-2 infection [10] have been extensively described [11]. Nevertheless, with the increasing use of real-time reverse transcription–polymerase chain reaction [RT-PCR] tests for diagnosis in COVID-19 patients, chest CT has progressively evolved its role beyond an exclusively diagnostic tool. Evidence focusing on the possible correlation between chest Computed Tomography (CT) findings and inflammatory status in COVID-19 patients [12,13] is available. Furthermore, chest CT has already shown prognostic value in COVID-19 patients [14–16], like in other lung conditions, including interstitial lung disease, lung cancer and other viral infections [17–19]. Moreover, CT plays a central role in the detection and monitoring of COVID-19-related pulmonary and extra-pulmonary complications [20,21].

The aim of this study was to investigate the correlation between chest CT findings, assessed through four different scores, serum inflammatory markers, and the presence of CS in patients naive to corticosteroids or any immunosuppressive drug. Moreover, the prognostic value of chest CT scores and the laboratory findings in COVID-19 patients specifically treated with anti-IL-6 drugs have been assessed.

## 2. Materials and Methods

### 2.1. Study Population and Treatment for COVID-19

The study has a retrospective longitudinal design. The study protocol was approved by the local institutional Committee on Research Ethics (Protocol no 0024185/20) in compliance with the Declaration of Helsinki. All the patients provided informed consent. Enrolment criteria included: hospitalization from 15 March to 15 April 2020, diagnosis of COVID-19 pneumonia confirmed through real-time reverse transcriptase polymerase chain reaction assay (RT-PCR) of respiratory secretions obtained by nasopharyngeal swab, eligibility to TCZ or SAR treatments because of disease severity. A baseline chest CT within 48 h from anti-IL-6R therapy initiation also had to be available for each patient. Exclusion criteria were ongoing treatment with corticosteroid or other immunosuppressive drugs, hospitalization in the intensive care unit (ICU) at the time of the CT and the presence of motion artefacts hampering the assessment of the CT.

IL-6 blockers were added to a standardized drug regimen based on the scientific evidence available at the time of the enrolment and shared by a multidisciplinary COVID-19 Task Force of our Hospital (Protocol no 926/2020 approved by the local Ethics Committee). In the absence of contraindications, the standard regimen for COVID-19 pneumonia

patients was made up of oral antiviral treatment (lopinavir/ritonavir 400/100 mg BD or darunavir/ritonavir 800/100 mg OD), oral hydroxychloroquine (400 mg BD for the first day, followed by 200 mg BD) and oral or intravenous azithromycin (500 mg OD). Low-dose heparin could be added to prevent deep vein thrombosis according to risk-factor assessment, and low-dose corticosteroids could be added after ICU admission.

Anti-IL-6R therapy was added to the standard regimen in patients in the internal medicine ward considered at risk of sudden clinical deterioration, according to clinical judgment. These patients included those with respiratory distress with $PaO_2/FiO_2$ ratio < 300 or with considerable lung involvement on CT scans and high serum inflammatory biomarkers. The treatment was avoided in case of known untreated latent tuberculosis, chronic hepatitis B and C infection, history of complicated diverticulitis, active cancer, concurrent bacterial or fungal infection, severe neutropenia or thrombocytopenia.

TCZ was the first choice anti-IL-6R and was intravenously administered at a dosage of 8 mg/kg (up to 800 mg for a single infusion), while a second infusion—24–96 h apart—was eventually administered based on clinical response and condition. SAR was used as an alternative because of the growing demand for IL-6R blockers that led to the rapid exhaustion of TCZ stocks in the first COVID-19 wave in Italy. Since SAR takes some days to reach its maximum concentration when administered subcutaneously, the drug was given intravenously by mixing SAR prefilled syringes in 0.9% sodium chloride solution for intravenous use according to the Italian Medicines Agency (AIFA) protocol. SAR was administered at the fixed dose of 400 mg, while a further dose of 400 mg was given after 24–96 h if the patient failed to improve.

### 2.2. Clinical, Laboratory Variables Collection and Computed Tomography Acquisitions

The data on demographics, comorbidities, onset of symptoms, treatments, laboratory results, and clinical outcomes of the patients were obtained from the electronic medical records. Clinical and laboratory data of the patients were collected within 24 h before the first TCZ or SAR administration. Standard blood tests included complete blood count with differential inflammatory markers (including C-reactive protein—CRP-, ferritin, fibrinogen, and D-dimer), Troponin I, metabolic profile on a venous blood sample and pO2 on an arterial blood sample to compute the pO2/FiO2 ratio. Peripheral venous blood was also collected for the assessment of IL-6, IL-1, IL-8 and Tumor Necrosis Factor $\alpha$ (TNF-$\alpha$) plasma levels using ELISA assay (Multi-cytokine test for Ella Bio-Techne).

Laboratory results at baseline have been used to define the presence of CS as defined by published criteria in COVID-19 pneumonia [6] (see Supplementary Materials for extensive description).

ICU admission, death, and oxygen weaning were considered outcome measures, and the enrolled patients were followed up for a 30-day period starting with the first (or single) anti-IL-6 dose.

Patients included in the study underwent a baseline CT scan for diagnostic purposes based on clinical symptoms before the availability of RT-PCR results within 48 h from the initiation of anti-IL-6R therapy. All chest CTs were acquired on a multidetector CT unit (Revolution CT, General Electric Healthcare, Milwaukee, WI, USA) in a dedicated CT room for cases with confirmed or suspected SARS-CoV-2 infection. Images were acquired in inspiration breath hold, with the following parameters: tube voltage of 120 kVp, pitch value of 1.5 and gantry rotation time of 0.35 s. Automatic tube current modulation was systematically used, with a range of 120–500 mAs and a noise index of 23.54. Images were reconstructed at 1.25 mm slice thickness with 1.25 mm increment, using the standard and high-resolution kernel for the assessment of mediastinum and lung parenchyma, respectively. CT acquisitions were all performed without contrast medium administration.

### 2.3. Computed Tomography Image Interpretation

Since the beginning of the pandemic, many studies have been conducted to define the best way to quantify the extension of CT lung alterations and to outline its potential

diagnostic and prognostic value, showing promising results. Indeed, several approaches have been reported in the literature to score CT scans in COVID-19 patients with two main differences. The first difference is related to the way the lung is divided, with some considering the five lobes [10,15] and other six zones: upper (above the carina), middle (below the carina up to the inferior pulmonary vein), and lower (below the inferior pulmonary vein) zones [22,23]. The second difference is represented by the way the score is calculated, with some considering only the extension of the lung abnormality [24,25] and other extension and types of CT findings (ground glass or consolidation) [22,26]. In order to assess the possible differences between these approaches, we decided to use more than one scoring system. All CT examinations were read by two thoracic radiologists (L.C. and G.C., 11 and 5 years of experience), who reached a decision in consensus. Lungs were divided into 5 lobes and 6 zones, as described above.

For each lobe and zone, the predominant CT finding was identified and graded as 1 for normal attenuation, 2 for ground-glass attenuation and 3 for consolidation. Each lobe or zone was also scored for extension of the affected lung parenchyma: 0 for unaffected lung, 1 for <25% abnormality, 2 for 25–50% abnormality, 3 for 50–75% abnormality, and 4 for >75% abnormality. From these assessments, four scores were derived; two scores were calculated by summing the extension of the affected lung, respective of the five lobes (S20 score) or of the six zones (S24 score) and two scores were calculated by multiplying the extension of disease by the predominant CT findings for each of the five lobes (S60 score) or of the six lung zones (S72 score). The axial distribution of the CT findings was classified as central, peripheral, or random. In addition to the semi-quantitative scores, the presence of other findings was evaluated, including the presence of centrilobular nodules, cavitation, crazy paving, airway abnormalities, pleural effusion, lymph node enlargement and the presence of underlying lung disease.

*2.4. Statistical Analysis*

Categorical variables were reported as numbers and percentages. Continuous variables were reported as mean ± SD (standard deviation) or median with interquartile range (IQR), according to the normality of the data. Analysis of categorical variables was performed with the chi-square test or Fisher's exact test, as appropriate. Comparisons between groups of continuous variables were performed with the Mann-Whitney U-test or *t*-test, according to the data distribution and homogeneity of variances. The linearity of the relationship between continuous variables was explored using Pearson's ($\rho$) or Spearman's coefficient (rs), as indicated. Univariate and multivariable Cox proportional hazards models were used to explore the associations between CT scores and the outcomes. CT scores and possible confounder variables were first screened in the univariate model and then included in the adjusted model if associated with a *p*-value $\leq 0.05$. The results were presented as hazard ratios (HRs) and 95% confidence intervals (CIs). The cut-off of CT scores and other continuous variables with the best ability to predict the 30-day outcomes according to the higher Youden Index of the Receiver Operating Characteristic (ROC) curve was defined when statistically feasible. Kaplan-Meier survival analysis was also conducted to assess the association of the established cut-off of CT scan severity with the time to ICU admission. A log-rank test was conducted to determine if there were differences in the survival distribution for the different cut-offs of CT scan severity. Statistical significance was defined as a *p* < 0.05. Data were analyzed using SPSS Statistics v26.0 (IBM, Armonk, NY, USA).

### 3. Results

*3.1. Study Population Selection*

Four-hundred and thirty-five symptomatic patients were diagnosed with COVID-19 at the Emergency Department of our hospital from 15 March to 15 April 2020, and none of them were therefore vaccinated. Among them, 113 were treated with anti-IL-6R agents because they were at risk of or with ongoing clinical deterioration. Chest CT was available

within 48 h from anti-IL-6R therapy initiation for 56 patients. Five patients were already hospitalized in ICU at the time of CT scan acquisition and were excluded from the study population since ICU admission was chosen as a prognostic outcome measure. The final study population was therefore made up of 51 patients, 26 treated with TCZ and 25 treated with SAR. Given that all the CT scans were considered of adequate quality, 51 patients were included in the final analysis (Figure 1).

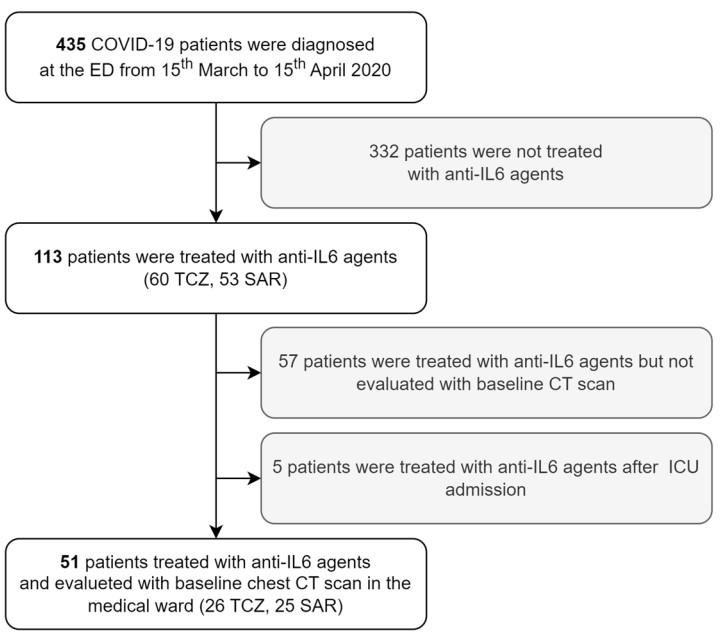

**Figure 1.** Patient selection process for the longitudinal analysis of COVID-19 outcome according to baseline chest CT scan. Abbreviations: CT computed tomography, COVID-19 coronavirus disease 2019, ED emergency department, TCZ tocilizumab, SAR sarilumab, IL interleukin, ICU intensive care unit.

The characteristics of the enrolled patients are summarized in Table 1. TCZ and SAR were administered twice to 15 (57.7%) and 8 (32%) patients, respectively. All the patients were treated with hydroxychloroquine and azithromycin, while 34 (66.7%) received a darunavir/ritonavir combination, 17 (33.3%) lopinavir/ritonavir combination, and 32 (65.3%) anticoagulation.

**Table 1.** Characteristics of the enrolled patients according to the administered anti-IL-6R treatment.

| | All Patients<br>*n* = 51 | TCZ<br>*n* = 26 | SAR<br>*n* = 25 | *p* |
|---|---|---|---|---|
| Male, *n* (%) | 41 (80.4) | 20 (76.9) | 21 (84.4) | 0.53 |
| Age, years, mean ± SD | 62.6 ± 12.5 | 60.8 ± 12.3 | 64.5 ± 12.7 | 0.29 |
| Disease duration, days, mean ± SD | 12.9 ± 6.0 | 14.4 ± 7.3 | 11.4 ± 3.0 | 0.74 |
| Diabetes, *n* (%) | 14 (27.5) | 7 (26.9) | 7 (28.0) | 0.93 |
| Coronary heart disease, *n* (%) | 10 (19.6) | 3 (11.5) | 7 (28.0) | 0.14 |
| Active cancer, *n* (%) | 0 (0) | 0 (0) | 0 (0) | - |
| COPD, *n* (%) | 0 (0) | 0 (0) | 0 (0) | - |
| pO2/FiO2, mean ± SD | 207 ± 79 | 231 ± 91 | 186 ± 61 | 0.05 |

**Table 1.** *Cont.*

| | All Patients<br>*n* = 51 | TCZ<br>*n* = 26 | SAR<br>*n* = 25 | *p* |
|---|---|---|---|---|
| CRP, mg/dL, mean ± SD | 124 ± 86 | 100 ± 94 | 134 ± 79 | 0.20 |
| Ferritin, mg/dL, median (IQR) | 656 (482–1464) | 646 (537–1290) | 1089 (425–1962) | 0.81 |
| Albumin, g/dL, mean ± SD | 3.0 ± 0.5 | 3.0 ± 0.4 | 3.0 ± 0.5 | 0.78 |
| Lymphocytes, %, median (IQR) | 12.4 (7.5–22.1) | 9.6 (7.0–20.6) | 14.3 (8.4–23.1) | 0.39 |
| Neutrophiles, n/mcl, mean ± SD | 5621 ± 3058 | 6684 ± 3267 | 4808 ± 2707 | 0.10 |
| ALT, mg/dL, median (IQR) | 32 (23–44) | 32 (23–44) | 31 (24–43) | 0.99 |
| AST, mg/dL, median (IQR) | 24 (22–42) | 24 (23–85) | 28 (19–28) | 0.91 |
| Dimers, mg/dL, median (IQR) | 1523 (718–3683) | 1083 (435–5840) | 1525 (1039–3359) | 0.28 |
| LDH, mg/dL, mean ± SD | 360 ± 111 | 347 ± 113 | 375 ± 111 | 0.47 |
| Troponin, ng/mL, median (IQR) | 0.07 (0.03–0.34) | 0.08 (0.04–0.27) | 0.04 (0.03–0.39) | 0.76 |
| Anion gap, mEq/L, mean ± SD | 14.5 ± 3.1 | 16.8 ± 3.3 | 12.5 ± 0.8 | 0.01 |
| Chloride, mEq/L, mean ± SD | 103.0 ± 5.4 | 102.4 ± 6.5 | 103.4 ± 4.7 | 0.70 |
| Potassium, mEq/L, mean ± SD | 3.8 ± 0.4 | 3.9 ± 0.5 | 3.8 ± 0.4 | 0.64 |
| Creatinine, mg/dL, median (IQR) | 0.9 (0.9–1.2) | 1.0 (0.8–1.2) | 0.9 (0.8–1.2) | 0.14 |
| BUN/creatinine ratio, median (IQR) | 18.2 (14.6–24.7) | 18.3 (15.1–25.6) | 17.9 (13.6–24.8) | 0.65 |
| TCZ/SAR, *n* (%) | 26 (51.0)/25 (49.0) | - | - | - |
| Hydroxychloroquine, *n* (%) | 51 (100.0) | 25 (100.0) | 26 (100.0) | - |
| Azithromycin, *n* (%) | 51 (100.0) | 25 (100.0) | 26 (100.0) | - |
| Darunavir/ritonavir, *n* (%) | 34 (66.7) | 15 (57.7) | 19 (76.0) | 0.17 |
| Lopinavir/ritonavir, *n* (%) | 17 (33.3) | 11 (42.3) | 6 (24.0) | 0.17 |
| LMWH, *n* (%) | 32 (65.3) | 12 (50.0) | 20 (80.0) | 0.03 |

Abbreviations: SD standard deviation, IQR interquartile range, COPD chronic obstructive pulmonary disease, CRP C-reactive protein, ALT alanine aminotransferase, AST aspartate aminotransferase, IL-6R Inteurlikin-6 receptor, LDH lactate dehydrogenase, BUN blood urea nitrogen, TCZ Tocilizumab, SAR sarilumab, LMWH low molecular weight heparin.

### 3.2. CT Scores, Clinical Characteristics, and Laboratory Variables

At baseline CT, the difference between the upper, middle and lower zones in terms of extension of lung involvement was not statistically significant ($p > 0.05$), reflecting a lack of craniocaudal predilection of the findings (Figure 2). Ground glass was the prevalent imaging finding compared to consolidation ($p < 0.05$) (Figure 3). The right middle zone and both lower zones showed a higher frequency of consolidation ($p < 0.05$). A summary of CT findings is reported in Table 2.

The mean value of each CT score described above was: 8.6 ± 3.6 for the S20 score, 10.4 ± 4.4 for the S24 score, 19.6 ± 7.0 for the S60 score, 23.6 ± 8.5 for the S72 score. The correlations between CT scores, disease characteristics and laboratory variables are reported in Table 3. No correlation was found between disease duration and CT scores. All the considered CT scores showed a negative correlation with pulmonary function, expressed by PaO2/FiO2 ratio, and a positive one with CRP, IL-6, IL-8, and TNF-α. The positive correlation of IL-1 reached statistical significance only for S60 and S72. The statistical significance was also preserved after Log-transformation for IL-6 and IL-8.

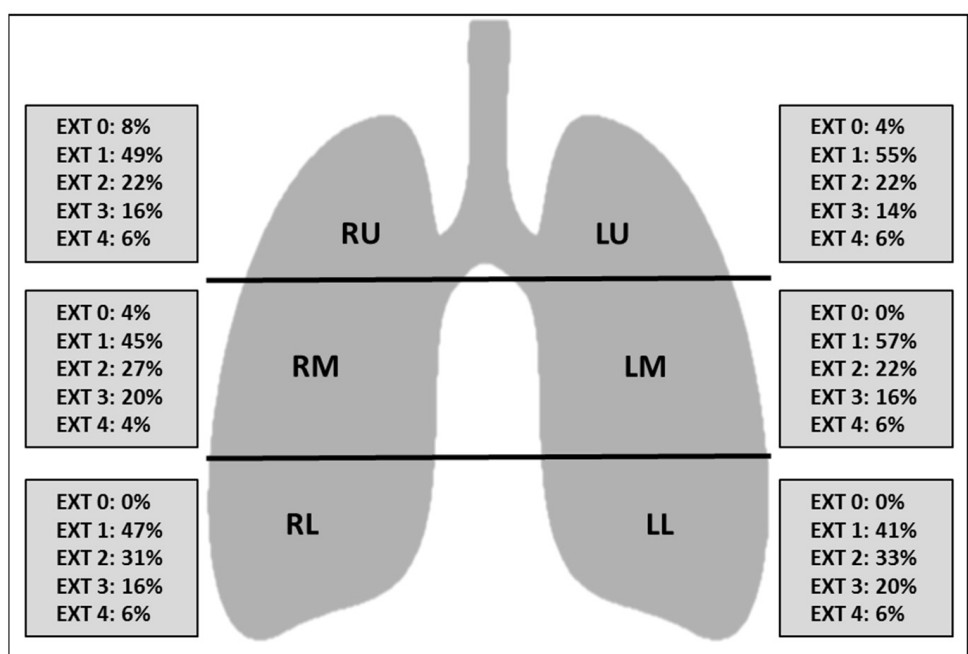

**Figure 2.** Schematic representation of the extension of lung abnormalities divided per lung zones. Numbers represent the percentage of patients. Abbreviations: EXT extension, RU right upper zone, LU left upper zone, RM right middle zone, LM left middle zone, RL right lower zone, LL left lower zone, EXT 0 no involvement, EXT 1 < 25% abnormality, EXT 2 25–50% abnormality, EXT 3 50–75% abnormality, EXT 4 > 75% abnormality.

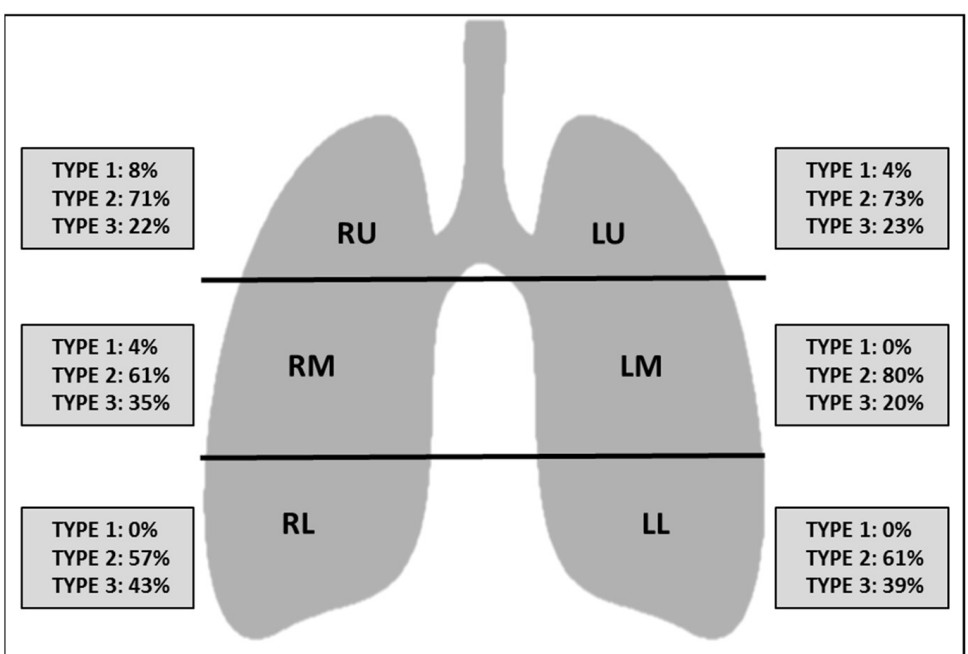

**Figure 3.** Schematic representation of the type of CT findings divided per lung zones. Numbers represent the percentage of patients. Abbreviations: RU right upper zone, LU left upper zone, RM right middle zone, LM left middle zone, RL right lower zone, LL left lower zone, Type 1 normal attenuation, Type 2 ground glass attenuation, Type 3 consolidation.

**Table 2.** Frequency of CT Findings.

| CT Findings | | *n.* (Percentage) |
|---|---|---|
| Centrilobular nodules | | 0 (0) |
| Pleural effusion | Right | 0 (0) |
| | Left | 5 (9.8) |
| | Bilateral | 15 (29.4) |
| Cavitation | | 0 (0) |
| Lymph node enlargement (lymph node sized ≥10 mm in short-axis dimension) | | 17 (33.3) |
| Airways abnormalities | Bronchial wall thickening | 2 (3.9) |
| | Bronchiectasis | 9 (17.6) |
| | Endoluminal secretions | 0 (0) |
| Axial distribution | Random | 30 (58.8) |
| | Central | 0 (0) |
| | Peripheral | 21 (41.2) |
| Crazy Paving | | 10 (19.6) |
| Underlying disease | Fibrosis | 0 (0) |
| | Emphysema | 7 (13.7) |

Abbreviations: CT computed tomography.

**Table 3.** Correlation of CT score with clinical and laboratory variables.

| | | S20 | | S24 | | S60 | | S72 | |
|---|---|---|---|---|---|---|---|---|---|
| Male | ρ (*p*) | 0.16 | (0.28) | 0.17 | (0.23) | 0.18 | (0.22) | 0.20 | (0.15) |
| Age | ρ (*p*) | 0.16 | (0.27) | 0.15 | (0.31) | 0.16 | (0.30) | 0.12 | (0.39) |
| Disease duration | $r_s$ (*p*) | 0.12 | (0.41) | −0.12 | (0.40) | −0.05 | (0.73) | −0.04 | (0.78) |
| Diabetes | ρ (*p*) | 0.00 | (0.82) | −0.04 | (0.79) | −0.04 | (0.77) | −0.06 | (0.67) |
| Coronary heart disease | ρ (*p*) | 0.27 | (0.06) | 0.24 | (0.09) | 0.18 | (0.20) | 0.15 | (0.29) |
| pO2/FiO2 | ρ (*p*) | −0.29 | (0.04) | −0.33 | (0.02) | −0.32 | (0.03) | −0.36 | (0.01) |
| CRP | ρ (*p*) | 0.44 | (<0.01) | 0.45 | (<0.01) | 0.34 | (0.01) | 0.37 | (<0.01) |
| Dimers | $r_s$ (*p*) | 0.17 | (0.31) | 0.19 | (0.24) | 0.23 | (0.16) | 0.23 | (0.16) |
| Ferritin | $r_s$ (*p*) | 0.23 | (0.31) | 0.23 | (0.32) | 0.19 | (0.41) | 0.12 | (0.41) |
| IL-6 | $r_s$ (*p*) | 0.59 | (<0.01) | 0.60 | (<0.01) | 0.55 | (<0.01) | 0.57 | (<0.01) |
| IL-6 * | ρ (*p*) | 0.54 | (<0.01) | 0.55 | (<0.01) | 0.47 | (<0.01) | 0.49 | (<0.01) |
| IL-8 | $r_s$ (*p*) | 0.53 | (<0.01) | 0.52 | (<0.01) | 0.40 | (<0.01) | 0.56 | (<0.01) |
| IL-8 * | ρ (*p*) | 0.45 | (<0.01) | 0.45 | (<0.01) | 0.35 | (0.02) | 0.33 | (0.02) |
| IL-1 | $r_s$ (*p*) | 0.28 | (0.07) | 0.28 | (0.07) | 0.35 | (0.03) | 0.33 | (0.03) |
| IL-1 * | ρ (*p*) | 0.17 | (0.29) | 0.13 | (0.40) | 0.19 | (0.22) | 0.15 | (0.33) |
| TNF-α | ρ (*p*) | 0.36 | (0.02) | 0.35 | (0.02) | 0.36 | (0.03) | 0.38 | (0.01) |
| Cytokine storm | ρ (*p*) | 0.39 | (0.05) | 0.39 | (0.05) | 0.31 | (0.03) | 0.32 | (0.02) |

* Log-transformed. Abbreviations: CT computed tomography, ρ Pearson's coefficient, rs Spearman's coefficient, PaO2:FiO2 arterial oxygen partial pressure to fractional inspired oxygen ratio, CRP C-reactive protein, IL interleukin, TNF tumor necrosis factor. Statistically significant correlations are highlighted in grey.

According to the selected criteria, six patients (11.7%) had a CS condition. All the CT scores statistically differ in patients with and without a COVID-19 CS. A ROC analysis was run to compare the ability of different CT scores to predict the presence of CS, and the S20 score showed the cut-off that best predicts the presence of CS (Youden's index 0.533). In particular, a cut-off of 11.5 has shown a sensitivity of 67% and a specificity of 87% in predicting an associate CS during untreated COVID-19 pneumonia (AUC 0.785, *p* = 0.034). Full ROC analysis is reported in Table 4.

**Table 4.** ROC analysis showing the predicting power for CS of four CT scoring systems.

| | Area | IC 95% | *p* |
|---|---|---|---|
| S20 | 0.785 | 0.579–0.991 | 0.02 |
| S24 | 0.776 | 0.569–0.983 | 0.03 |
| S60 | 0.739 | 0.538–0.939 | 0.06 |
| S72 | 0.746 | 0.559–0.933 | 0.05 |

Abbreviations: ROC Receiver Operating Characteristic, CS Cytokine Storm, CT Computed Tomography, IC interval of confidence.

### 3.3. Predictors of ICU Admission and Oxygen Weaning

During the 30-day follow-up, 11 patients (21.6%) were admitted to ICU and of these, two died because of COVID-19 or related complications. At the end of the same period, 43 patients (84.3%) had interrupted oxygen. Results of univariate and adjusted Cox regression models for ICU admission and oxygen weaning are shown in Table 5. In univariate analysis, higher S20, S24, S60, and S72 and Log-transformed IL-6 and IL-8 were risk factors for early ICU admission. In multivariate analysis, S24 was notably the only CT score that was associated with ICU admission when adjusted for Log-transformed IL-6 and IL-8. Moreover, in univariate analysis, lower S20, S24, S60, and S72, $pO_2/FiO_2$, TNF-$\alpha$ and Log-transformed IL-6 and IL-8, were associated with early oxygen weaning. However, in multivariate analysis, any CT score independently predicts early oxygen weaning when the model was adjusted for the other laboratory-associated variables. Of note, the presence of CS was not associated with later oxygen weaning or a higher risk of ICU admission.

**Table 5.** Cox proportional-hazard regression analysis of ICU admission and oxygen weaning risks.

| | ICU Admission | | | | | | Oxygen Weaning | | | | | |
|---|---|---|---|---|---|---|---|---|---|---|---|---|
| | Unadjusted | | | Adjusted † | | | Unadjusted | | | Adjusted ‡ | | |
| | HR | CI 95% | *p* | HR | CI 95% | *p* | HR | CI 95% | *p* | HR | CI 95% | *p* |
| S20 | 1.24 | 1.09–1.42 | <0.01 | 1.17 | 0.99–1.37 | 0.06 | 0.86 | 0.78–0.95 | 0.04 | 0.97 | 0.86–1.09 | 0.63 |
| S24 | 1.21 | 1.08–1.35 | <0.01 | 1.16 | 1.01–1.33 | 0.04 | 0.88 | 0.82–0.96 | <0.01 | 0.97 | 0.88–1.07 | 0.52 |
| S60 | 1.09 | 1.02–1.18 | 0.02 | 1.06 | 0.97–1.16 | 0.21 | 0.94 | 0.89–0.98 | <0.01 | 1.00 | 0.95–1.05 | 0.99 |
| S72 | 1.08 | 1.02–1.15 | 0.01 | 1.06 | 0.99–1.14 | 0.12 | 0.95 | 0.91–0.99 | <0.01 | 0.99 | 0.95–1.04 | 0.77 |
| Age | 0.98 | 0.93–1.02 | 0.28 | | | | 0.98 | 0.96–1.01 | 0.18 | | | |
| Male | 2.59 | 0.33–20.25 | 0.36 | | | | 0.90 | 0.43–1.88 | 0.77 | | | |
| Disease duration | 0.85 | 0.51–1.42 | 0.53 | | | | 1.07 | 0.86–1.33 | 0.56 | | | |
| CRP | 1.00 | 1.00–1.01 | 0.23 | | | | 1.00 | 0.99–1.00 | 0.09 | | | |
| pO2/FiO2 | 0.99 | 0.98–1.00 | 0.20 | | | | 1.01 | 1.00–1.01 | <0.01 | | | |
| Cytokine storm | 1.89 | 0.41–8.78 | 0.42 | | | | 0.17 | 0.04–0.72 | 0.16 | | | |
| IL-6 * | 2.13 | 1.12–4.07 | 0.02 | | | | 0.69 | 0.54–0.88 | <0.02 | | | |
| IL-8 * | 2.10 | 1.14–3.88 | 0.02 | | | | 0.60 | 0.39–0.94 | 0.03 | | | |
| IL-1 * | 2.71 | 0.78–9.43 | 0.12 | | | | 0.93 | 0.49–1.76 | 0.82 | | | |
| TNF-$\alpha$ | 0.999 | 0.92–1.09 | 0.98 | | | | 0.94 | 0.90–0.99 | 0.02 | | | |
| TCZ/SAR | 0.851 | 0.26–2.79 | 0.79 | | | | 1.12 | 0.613–2.03 | 0.72 | | | |

† Adjusted for Log IL-6 and Log IL-8; ‡ Adjusted for presence of Cytokine storm, $pO_2/FiO_2$, Log IL-6, Log IL-8 and TNF-$\alpha$; * Log-transformed. Abbreviations: ICU intensive care unit, HR Hazard Ratio, CI confidence interval, LDH lactate dehydrogenases, IL interleukin, TNF tumor necrosis factor, TCZ tocilizumab, SAR sarilumab. Statistically significant associations are highlighted in grey.

Given its relationship with ICU admission risk, the S24 was chosen to define a cut-off of CT score with a possible practical clinical utility. The S24 cut-off of 14.5 had the best ability to predict admission to ICU according to the higher Youden Index of the ROC curve (AUC 0.741 IC 95% 0.562–0.920; *p* = 0.015). In particular, this cut-off was associated with a sensitivity of 92.5% and a specificity of 54.5%. The patients were therefore divided according to this defined cut-off of S24, and Kaplan-Meier survival analysis was conducted to compare the rate of time to ICU admission and time of oxygen weaning into these two groups. Patients with S24 ≥ 14.5 (15.7%) had a shorter median time to ICU admission

(12.5 vs. 22.6 days, Log Rank $\chi^2$ = 13.091, $p < 0.01$) and a longer median time of oxygen weaning (22.8 vs. 12.0 days, Log Rank $\chi^2$ = 6.832, $p < 0.01$) (Figure 4).

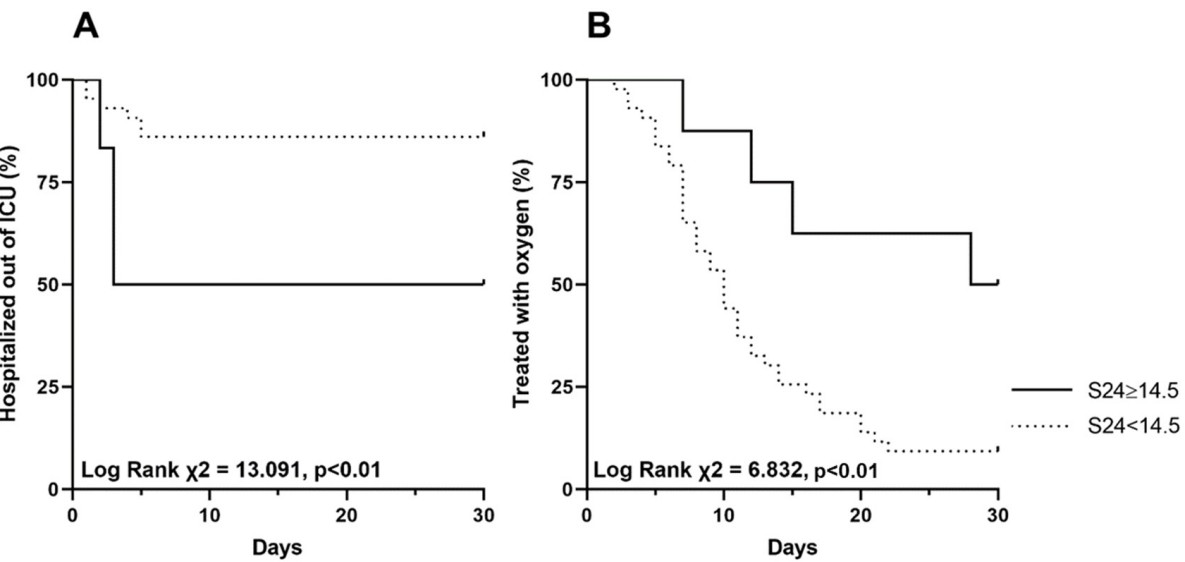

**Figure 4.** Kaplan-Meier graphs of time (days) to ICU admission (**A**) and time (days) of oxygen weaning (**B**) by the cut-off of 14.5 for CT scan score S24 in patients treated with anti-IL-6. Abbreviations: ICU intensive care unit, CT computed tomography, IL interleukin.

## 4. Discussion

An excessive inflammatory response with elevated serum proinflammatory cytokines and widespread CT scan abnormalities have been associated with critical and fatal SARS-CoV-2 infections [12,15,16,27,28]. In critically ill patients, the overactivated immune system is unable to eradicate the virus facilitating direct virus-mediated tissue damage [28,29], but also the inflammatory response induced by SARS-CoV-2 infection has been addressed as a cause of COVID-19 lung damage [30–32].

In our study, we explored the relationship between baseline CT findings and inflammatory state and compared the prognostic value of a comprehensive laboratory evaluation of systemic inflammation with CT scan scores in anti-IL-6 treated patients that during the first pandemic wave represented in our high-risk hospital patients. Four different scoring methods for CT were also applied and compared.

Our first observation was an association of CS with all the CT scores and a positive correlation of all the scores with specific cytokines, IL-6, IL-1, IL-8, and TNF-α. Notably, this observation was not biased by corticosteroid treatment as in other COVID-19 cohorts reported in the literature [9]. This finding supports the concept that lung damage relates to cytokine production in COVID-19 [33].

Regarding the prognostic assessment, we observed that in hospitalized patients with progressive respiratory worsening, higher CT scores were associated with earlier ICU admission and longer time for oxygen weaning, regardless of the scoring system used and despite all patients being treated with an analogue therapeutic approach. We reported an association of IL-6 and IL-8 levels with ICU admission and of IL-6, IL-8, and TNF-α with the oxygen weaning time. On multivariate analysis, only the S24 score preserved its prognostic value in the model adjusted for IL-6 and IL-8. In our cohort of severe COVID-19 patients, the prognostic value of CT scores seems, therefore, greater than basic clinical risk factors such as age, gender, CRP, or $PaO_2/FiO_2$ and comparable to laboratory biomarkers, such as IL-6 and IL-8. In particular, S24 was independently associated with ICU admission, indicating that a CT scan could improve prognostic stratification in hospitalized COVID-19 patients with progressive respiratory worsening and further treated with anti-IL-6 agents. It is intriguing to observe that the S24 cut-off identified for high-risk patients was doubled

compared to that identified in cohorts of patients not exposed to TCZ or SAR [34,35]. These data suggest that in patients treated with anti-IL6, a poor prognosis is associated with more severe parenchymal involvement.

It should be noted that we did not recognize any prognostic value for the presence of CS. A possible explanation for the lack of prognostic value of CS in our study might be due to the criteria used for its diagnosis. The proposed criteria of CS require ground glass opacity to be present at chest CT, regardless of its extension [6]. Since the inflammatory response is a pivotal factor but not the only factor in determining the severity of the disease, stricter radiological inclusion criteria could improve the prognostic value of the CS. The application of a cut-off value for the extension of lung abnormalities on CT could represent a way to quantify the extension of direct virus-mediated pulmonary damage independently of the systemic inflammatory biomarker response.

Different scoring methods have been used to evaluate CT scans in COVID-19 patients. In order to assess the possible differences between the several approaches, we tested more than one scoring system. Notably, the correlations of CT scores with inflammatory markers were similar for all the scoring systems without prominence for the pattern-adjusted scores (S60, S72). It can therefore be inferred that the correlation of imaging abnormalities with an inflammatory response is primarily extension-related rather than pattern-related in COVID-19 patients. Similarly, even though all the tested scores performed as prognostic factors, the S24 was the one providing the best performance in terms of prognostic assessment. Thus, our results highlight that CT extension of the affected lung parenchyma, regardless of appearance, is the main factor correlating with clinical impairment. Moreover, quantifying the extension of disease on six zones (S24) rather than on five lobes (S20) can perform better in evaluating the viral infection damage, which does not usually respect lobar margins.

Our study has some limitations. First, this is a single-center Italian study with a relatively small number of patients, and this reflects the difficulties in findings patients with similar clinical status and naive to corticosteroids and immunosuppressants. Nevertheless, it can be highlighted that thanks to the global vaccination campaign, the development of SARS-CoV-2 specific antiviral agents, and the widespread early use of corticosteroids in disease management, similar data about the infection behavior in patients naïve to all these interventions will probably not be available in the future. Second, we could not provide a control group of patients because, in our institution, during the first pandemic wave, all patients received treatment with anti-IL-6 when their respiratory conditions were worsening [36,37]. Third, we could not exclude with certainty the possible coexistence of other infections apart from SARS-CoV-2. Finally, we included only patients of the first wave of the pandemic, while other variants of the virus have been responsible for other waves. This could potentially affect the generalization of the results. However, recent studies have found no significant difference regarding CT pattern prevalence in different waves, and radiologic patterns in our study population were typical for COVID-19 [38,39].

## 5. Conclusions

In conclusion, our study on high-risk respiratory worsening in hospitalized patients naïve to corticosteroids demonstrated that extension of lung damage on CT is associated with inflammatory cytokine release but is also an independent prognostic factor for ICU admission and oxygen weaning. CT could represent a further tool to implement prognostic stratification in an attempt to optimize resource allocation. A better understanding of the respective role of inflammatory response, assessed through clinical and laboratory tests, and of direct virus-mediated tissue damage, assessed through the extension of lung damage at CT in the evolution of the disease, in particular in severe cases requiring hospitalization, will be helpful for tailoring COVID-19 treatments.

**Supplementary Materials:** The following supporting information can be downloaded at: https://www.mdpi.com/article/10.3390/tomography9030080/s1.

**Author Contributions:** Conceptualization, methodology, L.C., E.D.L., S.L.B. and A.R.L.; data curation, L.C., E.D.L. and A.I.; writing—original draft preparation, L.C. and E.D.L.; writing—review and editing, supervision, L.C., E.D.L., G.C., R.D., A.I., F.C., A.D.C., A.F., E.G., R.M., L.N., M.A.D., S.L.B. and A.R.L. All authors have read and agreed to the published version of the manuscript.

**Funding:** This research received no external funding.

**Institutional Review Board Statement:** The study was conducted in accordance with the Declaration of Helsinki and approved by the Institutional Committee on Research Ethics of Fondazione Policlinico universitario A. Gemelli (Protocol no 0024185/20).

**Informed Consent Statement:** Informed consent was obtained from all subjects involved in the study.

**Data Availability Statement:** Data are available upon request.

**Acknowledgments:** We thank Gemelli-Against-Covid Group: V. Abbate, N. Acampora, G. Addolorato, F. Agostini, M.E. Ainora, K. Akacha, E. Amato, F. Andreani, G. An- driollo, M.G. Annetta, B.E. Annicchiarico, M. Antonelli, G. Antonucci, G.M. Anzellotti, A. Armuzzi, F. Baldi, I. Barattucci, C. Barillaro, F. Barone, R.D.A. Bellantone, A. Bellieni, G. Bello, A. Benicchi, F. Benvenuto, L. Berardini, F. Berloco, R. Bernabei, A. Bianchi, D.G. Biasucci, L.M. Biasucci, S. Bibbò, A. Bini, A. Bisanti, F. Biscetti, M.G. Bocci, N. Bonadia, F. Bongiovanni, A. Borghetti, G. Bosco, S. Bosello, V. Bove, G. Bramato, V. Brandi, T. Bruni, C. Bruno, D. Bruno, M.C. Bungaro, A. Buonomo, L. Burzo, A. Calabrese, M.R. Calvello, C. Cambise, G. Cammà, M. Candelli, G. Canistro, A. Cantanale, G. Capalbo, L. Capaldi, E. Capone, E. Ca- pristo, L. Carbone, S. Cardone, S. Carelli, A. Carfì, A. Carnicelli, C. Caruso, F.A. Casciaro, L. Catalano, P. Cat- tani, R. Cauda, A.L. Cecchini, L. Cerrito, M. Cesarano, A. Chiarito, R. Cianci, S. Cicchinelli, A. Ciccullo, M. Cicetti, F. Ciciarello, A. Cingolani, M.C. Cipriani, M.L. Consalvo, G. Coppola, G.M. Corbo, A. Corsello, F. Cos- tante, M. Costanzi, M. Covino, D. Crupi, S.L. Cutuli, S. D'Addio, A. D'Alessandro, M.E. D'Alfonso, E. D'Ange- lo, F. D'Aversa, F. Damiano, G.M. De Berardinis, T. De Cunzo, D.K. De Gaetano, G. De Luca, G. De Matteis, G. De Pascale, P. De Santis, M. De Siena, F. De Vito, V. Del Gatto, P. Del Giacomo, F. Del Zompo, A.M. Dell'Anna, D. Della Polla, L. Di Gialleonardo, S. Di Giambenedetto, R. Di Luca, L. Di Maurizio, M. Di Muro, A. Dusina, D. Eleuteri, A. Esperide, D. Fachechi, D. Faliero, C. Fal- siroli, M. Fantoni, A. Fedele, D. Feliciani, C. Ferrante, G. Ferrone, R. Festa, M.C. Fiore, A. Flex, E. Forte, A. Francesconi, L. Franza, B. Funaro, M. Fuorlo, D. Fusco, M. Gabrielli, E. Gaetani, C. Galletta, A. Gallo, G. Gam- bassi, M. Garcovich, I. Gasparrini, S. Gelli, A. Giampi- etro, L. Gigante, G. Giuliano, G. Giuliano, B. Giupponi, E. Gremese, D.L. Grieco, M. Guerrera, V. Guglielmi, C. Guidone, A. Gullì, A. Iaconelli, A. Iafrati, G. Ianiro, A. Iaquinta, M. Impagnatiello, R. Inchingolo, E. Intini, R. Iorio, I.M. Izzi, T. Jovanovic, C. Kadhim, R. La Mac- chia, F. Landi, G. Landi, R. Landi, R. Landolfi, M. Leo, P.M. Leone, L. Levantesi, A. Liguori, R. Liperoti, M.M. Lizzio, M.R. Lo Monaco, P. Locantore, F. Lombardi, G. Lombardi, L. Lopetuso, V. Loria, A.R. Losito, B.P.L. Mothanje, F. Macagno, N. Macerola, G. Maggi, G. Mai- uro, F. Mancarella, F. Mangiola, A. Manno, D. Marchesi- ni, G.M. Maresca, G. Marrone, I. Martis, A.M. Martone, E. Marzetti, C. Mattana, M.V. Matteo, R. Maviglia, A. Mazzarella, C. Memoli, L. Miele, A. Migneco, I. Migni- ni, A. Milani, D. Milardi, M. Montalto, G. Montemurro, F. Monti, L. Montini, T.C. Morena, V. Mora, C. Morretta, D. Moschese, C.A. Murace, M. Murdolo, M. Napoli, E. Nardella, G. Natalello, D. Natalini, S.M. Navarra, A. Nesci, A. Nicoletti, R. Nicoletti, T.F. Nicoletti, R. Nicolò, E.C. Nista, E. Nuzzo, M. Oggiano, V. Ojetti, F. Cosimo Pagano, G. Paiano, C. Pais, F. Pallavicini, A. Palombo, F. Paolillo, A. Papa, D. Papanice, L.G. Papparella, M. Para- tore, G. Parrinello, G. Pasciuto, P. Pasculli, G. Pecorini, S. Perniola, E. Pero, L. Petricca, M. Petrucci, C. Picarel- li, A. Piccioni, A. Piccolo, E. Piervincenzi, G. Pignataro, R. Pignataro, G. Pintaudi, L. Pisapia, M. Pizzoferrato, F. Pizzolante, R. Pola, C. Policola, M. Pompili, F. Pontecor- vi, V. Pontecorvi, F.R. Ponziani, V. Popolla, E. Porced- du, A. Porfidia, L.M. Porro, A. Potenza, F. Pozzana, G.Privitera, D. Pugliese, G. Pulcini, S. Racco, F. Raffaelli, V. Ramunno, G.L. Rapaccini, L. Richeldi, E. Rinninella, S. Rocchi, B. Romanò, S. Romano, F. Rosa, L. Rossi, R. Rossi, E. Rossini, E. Rota, F. Rovedi, C. Rubino, G. Rumi, A. Russo, L. Sabia, A. Salerno, M. Sali, S. Salini, L. Salvatore, D. Samori, C. Sandroni, M. Sanguinetti, R. Santangelo, L. Santarelli, P. Santini, D. Santolamazza, A. Santoliquido, F. Santopaolo, M.C. Santoro, F. Sardeo, C. Sarnari, A. Saviano, L. Saviano, F. Scaldaferri, R. Scarascia, T. Schepis, F. Schiavello, G. Scoppettuolo, D. Sedda, F. Sessa, L. Sestito, C. Settanni, M. Siciliano, V. Siciliano, R. Sicuranza, B. Simeoni, J. Simonetti, A. Smargiassi, P.M. Soave, C. Sonnino, D. Staiti, C. Stella, L. Stella, E. Stival, E. Taddei, R. Talerico, E. Tamburel- lo, E. Tamburrini, E.S. Tanzarella, E. Tarascio, C. Tarli, A. Tersali, P. Tilli, J. Timpano, E. Torelli, F. Torrini, M. Tosato, A. Tosoni, L. Tricoli, M. Tritto, M. Tumbarello, A.M. Tummolo, M.S. Vallecoccia, F. Valletta, F. Varone, F. Vassalli, G. Ventura, L.

Verardi, L. Vetrone, E. Vis- conti, F. Visconti, A. Viviani, R. Zaccaria, C. Zaccone, C. Zanza, L. Zelano, L. Zileri Dal Verme, G. Zuccalà.

**Conflicts of Interest:** The authors declare no conflict of interest.

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
