# Peer review of "Extension of Lung Damage at Chest Computed Tomography in Severely Ill COVID-19 Patients Treated with Interleukin-6 Receptor Blockers Correlates with Inflammatory Cytokines Production and Prognosis"

_tomography, doi:10.3390/tomography9030080_

Round 1

Reviewer 1 Report

Dear Authors

I would like to thank you for the opportunity of reviewing this interesting paper that is focused on a very remarkable and challenging topic that is a lively argument also in daily clinical practice. Although it has been more than 2 years since the first outbreak, the coronavirus disease 2019 (COVID-19) pandemic is still having a profound and devastating impact on global healthcare systems. COVID-19 is responsible for a respiratory disease whose broad spectrum of severity ranges from asymptomatic or mildly symptomatic infection to severe bilateral pneumonia, which may lead to acute respiratory distress syndrome (ARDS), requiring non-invasive or invasive mechanical ventilation and Intensive Care Unit (ICU) admission. Elevated inflammatory markers are associated with severe COVID-19 and some patients benefit from Interleukin (IL)-6 pathway inhibitors. Different CT scoring systems have shown a prognostic value in COVID-19, but not specifically in anti-IL-6-treated patients at high risk of respiratory failure. The present study aimed to explore the relationship between baseline CT findings and inflammatory condition and to evaluate the prognostic value of chest CT scores and laboratory findings in COVID-19 patients specifically treated with anti-IL-6.

This paper is pleasurable to read, although it suffers from some limitations that Authors can easily adjust in order to improve their review making it more eligible for this important Journal. Furthermore, the Authors can improve some sections of the paper, adding information and including other important references about this topic that, in my opinion, should be cited and discussed. 

Although the language used is quite appropriate, I (I am not a native English speaker) recommend to the Authors to obtain a certified native speaker with proficiencies in the scientific-medical field to complete properly this paper (if not yet done). Moreover, I recommend making a further revision of the manuscript to fix some small typing/language errors. 

The title is clear and direct. However, from a stylistic point of view, I believe it could be improved and more focused on the results. For example, “Extension of lung damage at chest computed tomography in severely ill COVID-19 patients treated with interleukin-6 receptor blockers correlates with inflammatory cytokines production

The Authors did not correctly report keywords from MeSH Browser. In particular, I believe there is an error: “Tomography” and “X-Ray Computed”. I suggest the check of all KW. 

The abstract is correct; however, I suggest adding the statistical results of the study, reporting the corresponding p-values.

Although the introduction fits the context of the study, many concepts could be more clearly explicated in an exhaustive introduction, which would help readers to become passionate about reading the paper and using it as a reference. 

In my opinion, it is very important to underline how the role of CT in COVID-19 patients has evolved during the pandemic, especially those who were severely ill. Given the progressively increased availability of RT-PCR, in fact, it was interesting to note that CT changed from a primarily diagnostic to a prognostic role during the pandemic. Now, chest CT is one of the main techniques to assess the severity of the pneumonic infection, allowing patients to be stratified into risk categories, estimate their prognosis, and help in medical decision making for hospitalization. Moreover, CT plays a pivotal role in the detection and monitoring of COVID-19-related complications, most of which are life-threatening, thus improving patient outcomes and overall survival. For example, it represents the best modality choice to indagate vascular alterations, both pulmonary and visceral, and can provide adequate information on the state of the gastrointestinal and renal systems, also allowing the evaluation of possible brain injuries. [Diagnostics (Basel). 2022 Mar 29;12(4):846. doi: 10.3390/diagnostics12040846][Radiographics. 2020 Nov-Dec;40(7):1848-1865. doi: 10.1148/rg.2020200159]. Please, cite both these articles and introduce these important aspects in this section.

Furthermore, I believe that a brief introduction regarding the available CT scores, especially those used in the present study, should be made.

In Material & Methods section, the study period was between 15th March to 15th April 2020, which corresponds to the first wave of the pandemic in Italy. Therefore, since several waves caused by different virus variants have been reported, the present results present an important bias. However, recent studies have demonstrated that this COVID-19 CT pattern prevalence according to the RSNA Expert Consensus Statement [Radiol Cardiothorac Imaging. 2020 Jun 11;2(3):e200213. doi: 10.1148/ryct.2020200213] did not statistically differ between the different waves of the pandemic and this new type of pneumonia tends to present itself with “typical” radiological features in the majority of cases [Emerg Radiol. 2021 Dec;28(6):1055-1061. doi: 10.1007/s10140-021-01937-y.]. Could please the Authors discuss this topic here or in the Discussion section, citing the aforementioned papers? Moreover, could the Authors report if the CT images used for the present analysis were from COVID-19 patients with “typical” appearance?

Furthermore, I believe the explanation of drugs administration submitted as a supplementary file (Supplementary Materials) should be included in the main text.  Please add a brief part regarding IL-6 inhibitors contraindications and merge 2.1 with 2.2. as well as 2.4 with 2.5. Finally, explain why some patients received tocilizumab while others sarilumab.

Both supplementary tables should be incorporated in the main text (Table 1 should be replaced by supplementary table 1).

Figure 2 and Figure 3 should be improved, adding a proper title for each one and replacing 0-4 with the involvement percentage (0%, < 25%, etc.) or the extended type of lesions (ground-glass, etc.).

Authors should detail more extensively the limitations of their study. For example, they should state that this is a single center study performed exclusively on Italian patients. Moreover, no other concomitant bacterial, fungal or viral infections were investigated and reported. This data is essential given that the probability of survival and ICU admission, as well as the levels of cytokines, could be influenced by the presence of any other microorganism. Indeed, it is known that ICU patients with COVID-19 often present with a concomitant fungal and bacterial infection [Chest. 2021 Aug;160(2):454-465. doi: 10.1016/j.chest.2021.04.002] [J Med Virol. 2022 May;94(5):1920-1925. doi: 10.1002/jmv.27548][ Diagnostics (Basel). 2022 Jul 3;12(7):1617. doi: 10.3390/diagnostics12071617]. If possible, therefore, the Authors should include these data in the analysis. If this data is not available, it is necessary to include this important limit at the end of the discussion.

Please add “Conclusion” before the conclusion section.

Finally, I think references should be reformatted as suggested by Tomography Author’s guidelines (Author 1, A.B.; Author 2, C.D. Title of the article. Abbreviated Journal Name YearVolume, page range)

Best regards, 

Author Response

Dear editor,

We thank you for your comments. We really appreciate your interest in our work.
As suggested, we did the revisions required.

Point 1:  Although the language used is quite appropriate, I (I am not a native English speaker) recommend to the Authors to obtain a certified native speaker with proficiencies in the scientific-medical field to complete properly this paper (if not yet done). Moreover, I recommend making a further revision of the manuscript to fix some small typing/language errors. 

Response 1: Thank you, we have checked the whole manuscript to correct typing/language errors.

Point 2: The title is clear and direct. However, from a stylistic point of view, I believe it could be improved and more focused on the results. For example, “Extension of lung damage at chest computed tomography in severely ill COVID-19 patients treated with interleukin-6 receptor blockers correlates with inflammatory cytokines production

Response 2: Thank you for your suggestion, we have modified the title according to your advice: “Extension of lung damage at chest computed tomography in severely ill COVID-19 patients treated with interleukin-6 receptor blockers correlates with inflammatory cytokines production and prognosis”.

Point 3: The Authors did not correctly report keywords from MeSH Browser. In particular, I believe there is an error: “Tomography” and “X-Ray Computed”. I suggest the check of all KW. 

Response 3: Thank you, we checked the keywords and corrected them as it follows: Computed Tomography, COVID-19, Prognosis, Cytokine Release Syndrome, Interleukin-6.

Point 4: The abstract is correct; however, I suggest adding the statistical results of the study, reporting the corresponding p-values.

Response 4: Thank you for your comment. We didn’t include all the results in order to respect the word limits for the abstract (200 words). However, as suggested, we added the p value of the correlation between disease extension assessed by the six-lung-zone CT score (S24) and ICU admission which we think is one of the more relevant. Including the p value for all the correlations between the several CT scores calculated and laboratory test (C-reactive protein (CRP), IL-6, IL-8, and Tumor Necrosis Factor α) would mean including a large number of p value and could make the abstract too long and confusing. Nevertheless, if the reviewers consider this important, we can include them.

Point 5:  Although the introduction fits the context of the study, many concepts could be more clearly explicated in an exhaustive introduction, which would help readers to become passionate about reading the paper and using it as a reference. 
In my opinion, it is very important to underline how the role of CT in COVID-19 patients has evolved during the pandemic, especially those who were severely ill. Given the progressively increased availability of RT-PCR, in fact, it was interesting to note that CT changed from a primarily diagnostic to a prognostic role during the pandemic. Now, chest CT is one of the main techniques to assess the severity of the pneumonic infection, allowing patients to be stratified into risk categories, estimate their prognosis, and help in medical decision making for hospitalization. Moreover, CT plays a pivotal role in the detection and monitoring of COVID-19-related complications, most of which are life-threatening, thus improving patient outcomes and overall survival. For example, it represents the best modality choice to indagate vascular alterations, both pulmonary and visceral, and can provide adequate information on the state of the gastrointestinal and renal systems, also allowing the evaluation of possible brain injuries. [Diagnostics (Basel). 2022 Mar 29;12(4):846. doi: 10.3390/diagnostics12040846][Radiographics. 2020 Nov-Dec;40(7):1848-1865. doi: 10.1148/rg.2020200159]. Please, cite both these articles and introduce these important aspects in this section.

Furthermore, I believe that a brief introduction regarding the available CT scores, especially those used in the present study, should be made.

Response 5: Thank you for your suggestions, we implemented the introduction with those articles. Regarding the CT scores, in the section ‘Computed tomography image interpretation’ of Materials and Methods we have inserted a brief introduction on CT findings quantification before explaining the two main types of scoring system found in literature and used in our study.

Point 6:  In Material & Methods section, the study period was between 15th March to 15th April 2020, which corresponds to the first wave of the pandemic in Italy. Therefore, since several waves caused by different virus variants have been reported, the present results present an important bias. However, recent studies have demonstrated that this COVID-19 CT pattern prevalence according to the RSNA Expert Consensus Statement [Radiol Cardiothorac Imaging. 2020 Jun 11;2(3):e200213. doi: 10.1148/ryct.2020200213] did not statistically differ between the different waves of the pandemic and this new type of pneumonia tends to present itself with “typical” radiological features in the majority of cases [Emerg Radiol. 2021 Dec;28(6):1055-1061. doi: 10.1007/s10140-021-01937-y.]. Could please the Authors discuss this topic here or in the Discussion section, citing the aforementioned papers? Moreover, could the Authors report if the CT images used for the present analysis were from COVID-19 patients with “typical” appearance?

Response 6: Thank you for your comment. We implemented the discussion with the article suggested. Our study population included only COVID-19 patients from the first wave of the pandemic and all of them showed typical CT pattern; we also added this information in the discussion as suggested.

Point 7: Furthermore, I believe the explanation of drugs administration submitted as a supplementary file (Supplementary Materials) should be included in the main text.  Please add a brief part regarding IL-6 inhibitors contraindications and merge 2.1 with 2.2. as well as 2.4 with 2.5. Finally, explain why some patients received tocilizumab while others sarilumab.

Both supplementary tables should be incorporated in the main text (Table 1 should be replaced by supplementary table 1).

Response 7: Thank you for your suggestions, we included the description of drug administration in the main text in Materials and Methods 2.1 and removed it from the supplementary material. 2.1 was merged with 2.2 and 2.4 with 2.5 as suggested. We also merged Table1 and supplementary Table 1 and added supplementary Table 2 in the main text as Table 4.

Point 8: Figure 2 and Figure 3 should be improved, adding a proper title for each one and replacing 0-4 with the involvement percentage (0%, < 25%, etc.) or the extended type of lesions (ground-glass, etc.).

Response 8: Thank you for your suggestions about the figures. We indicated CT findings extension as numbers from 0 to 4 and type of CT findings as numbers from 1 to 3 to show the results of the scoring performed by the radiologists as explained in the Materials and Methods section. The legenda in the figures’ caption clarifies the numbers meaning. If the reviewer thinks that making the changes required can improve the comprehension of the figures, we would be glad to make them. The only reason why we reported those numbers in the figure was to reproduce the results of the scoring as reported in the Materials and Methods section.

Point 9: Authors should detail more extensively the limitations of their study. For example, they should state that this is a single center study performed exclusively on Italian patients. Moreover, no other concomitant bacterial, fungal or viral infections were investigated and reported. This data is essential given that the probability of survival and ICU admission, as well as the levels of cytokines, could be influenced by the presence of any other microorganism. Indeed, it is known that ICU patients with COVID-19 often present with a concomitant fungal and bacterial infection [Chest. 2021 Aug;160(2):454-465. doi: 10.1016/j.chest.2021.04.002] [J Med Virol. 2022 May;94(5):1920-1925. doi: 10.1002/jmv.27548][ Diagnostics (Basel). 2022 Jul 3;12(7):1617. doi: 10.3390/diagnostics12071617]. If possible, therefore, the Authors should include these data in the analysis. If this data is not available, it is necessary to include this important limit at the end of the discussion.

Response 9: Thank you for your comment. We added the information requested as a limit of our study. This is indeed a single Italian center study and we couldn’t exclude with certainty the possible coexistence of other infections.

Point 10: Please add “Conclusion” before the conclusion section.

Response 10: Thank you, we added the term “Conclusion” before the conclusion section.

Point 11: Finally, I think references should be reformatted as suggested by Tomography Author’s guidelines (Author 1, A.B.; Author 2, C.D. Title of the article. Abbreviated Journal Name YearVolume, page range)

Response 11: Thank you, we modified references as suggested.

Best regards

Reviewer 2 Report

Dear Authors,

It was interesting for me to read your retrospective study on COVID-19 CT scoring systems with correlation to immune status of severe COVID-19 patients. I think it's a very important clinical and radiological matter, conclusions of the prognostic value of chest CT scores, especially S24,  are interesting, the research is well structured. 

I just think it would be important to note what percentage of patients were vaccinated if any (I presumed none at the time of spring 2020...?) but it should be clearly stated in the text.

Furthermore, for the better comprehension, in the text of CT scores (3.2) as well as the appropriate figures from this section, clear statements defining the time of the CT findings or CT scores should be noted, otherwise one may doubt what findings they represent (on initial scans or control scans?) - I would more often include "baseline CT images / scores" in the text.

The drawback of this study is a lack of control CT analysis after 1 month in surviving patients, that could better characterise the response to Anti-IL6 treatment and the follow-up of the pulmonary damage. The other drawback is the lack of a control research on a group of vaccinated patients in the future if it's difficult to find the control group without anti-Il6 treatment.

Nevertheless, the drawbacks don't lower the importance of the conclusions of this study and I therefore suggest to accept the article in this form, after some minor corrections (vide supra). 

Some minor English corrections that I have noticed:

- lines 52, Introduction, "response, defined AS A cytokine storm, "

Author Response

Dear editor,

We thank you for your useful comments. We really appreciate your interest in our work.
As suggested, we did the revisions required.

Point 1:  I just think it would be important to note what percentage of patients were vaccinated if any (I presumed none at the time of spring 2020...?) but it should be clearly stated in the text.

Response 1: Thank you for your suggestion, we clarified in section 3.1 that none of our patients were vaccinated at the time of the study.

Point 2: Furthermore, for the better comprehension, in the text of CT scores (3.2) as well as the appropriate figures from this section, clear statements defining the time of the CT findings or CT scores should be noted, otherwise one may doubt what findings they represent (on initial scans or control scans?) - I would more often include "baseline CT images / scores" in the text.

Response 2: Thank you for your comment. Our CT scans were all baseline CT, specifically the patients underwent CT scan within 48 hours from the initiation of anti-IL-6R therapy, as reported in section 2.1. To improve clarity of the text, we have specified in the text that they were baseline CT in section 2.1, 2.2 and 3.2.

Thank you again for your comments.

Best regards

Round 2

Reviewer 1 Report

Authors addressed raised points appropriately.